# Feasibility of Application of the Newly Developed Nano-Biomaterial, β-TCP/PDLLA, in Maxillofacial Reconstructive Surgery: A Pilot Rat Study

**DOI:** 10.3390/nano11020303

**Published:** 2021-01-25

**Authors:** Erina Toda, Yunpeng Bai, Jingjing Sha, Quang Ngoc Dong, Huy Xuan Ngo, Takashi Suyama, Kenichi Miyamoto, Yumi Matsuzaki, Takahiro Kanno

**Affiliations:** 1Department of Oral and Maxillofacial Surgery, Shimane University Faculty of Medicine, 89-1 Enya-Cho, Izumo 693-8501, Shimane, Japan; et1211@med.shimane-u.ac.jp (E.T.); xyywq@126.com (Y.B.); jsswjbnjw@gmail.com (J.S.); quangbeo87@gmail.com (Q.N.D.); ngoxuanhuy158@gmail.com (H.X.N.); 2Department of Life Science, Shimane University Faculty of Medicine, 89-1 Enya-Cho, Izumo 693-8501, Shimane, Japan; m169603@med.shimane-u.ac.jp (T.S.); miyaken@med.shimane-u.ac.jp (K.M.); matsuzak@med.shimane-u.ac.jp (Y.M.)

**Keywords:** β-TCP/PDLLA, Runx2, osteocalcin, leptin receptor, biocompatibility, osteoconductivity

## Abstract

This study was performed to examine the applicability of the newly developed nano-biocomposite, β-tricalcium phosphate (β-TCP)/u-HA/poly-d/l-lactide (PDLLA), to bone defects in the oral and maxillofacial area. This novel nano-biocomposite showed several advantages, including biocompatibility, biodegradability, and osteoconductivity. In addition, its optimal plasticity also allowed its utilization in irregular critical bone defect reconstructive surgery. Here, three different nano-biomaterials, i.e., β-TCP/PDLLA, β-TCP, and PDLLA, were implanted into critical bone defects in the right lateral mandible of 10-week-old Sprague–Dawley (SD) rats as bone graft substitutes. Micro-computed tomography (Micro-CT) and immunohistochemical staining for the osteogenesis biomarkers, Runx2, osteocalcin, and the leptin receptor, were performed to investigate and compare bone regeneration between the groups. Although the micro-CT results showed the highest bone mineral density (BMD) and bone volume to total volume (BV/TV) with β-TCP, immunohistochemical analysis indicated better osteogenesis-promoting ability of β-TCP/PDLLA, especially at an early stage of the bone healing process. These results confirmed that the novel nano-biocomposite, β-TCP/PDLLA, which has excellent biocompatibility, bioresorbability and bioactive/osteoconductivity, has the potential to become a next-generation biomaterial for use as a bone graft substitute in maxillofacial reconstructive surgery.

## 1. Introduction

With the increasing number of cases of bone loss caused by malignant tumors or trauma, autografting does not meet the demands of bone restoration, particularly in cases of large bone defects [1]. Autografts are often ineffective because of the irregular shape and size of the defect, as well as limited donor sources [2]. Defects in the maxillofacial region are especially difficult to treat due to the complex three-dimensional shape, aesthetic considerations, and importance of mastication and speech [3]. In routine bone graft surgery in the oral and maxillofacial surgery department, we frequently harvest autologous bone from iliac or fibula for transplantation into critical bone defects in the maxillofacial region, in combination with fixation of a titanium plate [4,5]. These conventional methods have a number of disadvantages, such as donor site morbidity and the requirement for secondary surgery [4]. Complete restoration of function in the area of the bone defect would significantly improve the quality of life of affected individuals and reduce the associated socioeconomic costs.

To achieve such restoration, various types of artificial bone graft substitutes have been developed to provide a suitable microenvironment for the growth and orientation of osteoblasts [6]. Since the early 1990s, poly-l-lactide (PLLA) has been used as the first-generation bioresorbable biomaterial for maxillofacial osteosynthesis [7,8]. Subsequently, copolymers of polyglycolic acid (PGA), PLLA, and poly-d-lactide (PDLA) were developed as second-generation biomaterials for use in place of pure PGA and PLLA [9]. However, due to insufficient intensity, foreign body reactions, and lack of osteoconductivity [10,11], the first- and second-generation biomaterials have very limited clinical utility [7]. These issues were addressed by third-generation bioresorbable/bioactive osteosynthetic nanomaterials consisting of forged composites of uncalcined and unsintered hydroxyapatite (u-HA)/PLLA [12,13]. Our previous study also demonstrated that the third-generation bioresorbable nanobiomaterials consisting of u-HA/poly-d/l-lactide (PDLLA) and u-HA/PLLA can provide good osteoconductivity and prompt bone regeneration [14,15,16,17]. In addition, the excellent plasticity of these biomaterials allows surgeons to reshape them easily to fit diverse bone defects intraoperatively [14,15,16].

Nevertheless, β-tricalcium phosphate (β-TCP) composed of purified inorganic tricalcium phosphate is still the predominant bone graft substitute used in clinical applications [18]. As it is a composite of calcium salt and phosphoric acid, β-TCP can undergo ion exchange with the original bone and accelerate the calcification of new bone [19]. In Japan, β-TCP is now covered by the national insurance system as a bone graft substitute [20]. β-TCP has a vast network of interconnected pores, which can directly benefit fibrovascular invasion and bone replacement [21]. However, this composite cannot be trimmed to the shape of the defect, and β-TCP implants with low stress levels show brittle fracture behavior in clinical applications [22]. To overcome the shortcomings of these materials, we added a suitable nano-organic composite PDLLA into β-TCP to confer greater plasticity [23]. PDLLA is an amorphous polymer composed of repeating units of l-lactic acid and d-lactic acid [15,16], which has been widely used in bioengineering due to its biocompatibility and biodegradability [7,16,18]. Here, a novel β-TCP/PDLLA nano-biocomposite was developed by Teijin Medical Technologies Co., Ltd. (Osaka, Japan) in an attempt to identify a fourth-generation bioresorbable biomaterial with better properties for promotion of bone regeneration.

To date, there have been few studies similar to this in vivo study, and the optimal ratio, porosity, and pore size of these types of composites have not been elucidated. This study focused on testing the biofeatures and bone regeneration capability of this novel nano-biomaterial in a rodent mandibular critical bone defect model, to identify an ideal biomaterial for clinical use, especially in the oral and maxillofacial regions.

## 2. Materials and Methods

### 2.1. Materials

Three different types of biocomposite were used in this study, i.e., β-TCP/PDLLA, pure β-TCP, and pure PDLLA. All of the composites were provided by Teijin (Teijin Medical Technologies, Osaka, Japan) The β-TCP/PDLLA biomaterials were composed of 70 wt% β-TCP and 30 wt% PDLLA matrix (d/l-lactide acid = 50/50 mol%).

The porosity calculated from the apparent density of β-TCP/PDLLA was the same as that of pure PDLLA, but lower than that of β-TCP (70% ± 5%, 70% ± 5%, and 77.5 ± 4.5%, respectively). Apparent density (*p_a_*) was calculated using the following equation [24]:$$p_{a},\%=\frac{W_{s}-W_{D}}{V}\cdot 100$$
where *W_S_* and *W**_D_* represent the saturated weight and dry weight of the biocomposite, respectively, and *V* is the exterior volume.

All of the materials were manufactured as cylinders 4 mm in diameter by 2 mm in thickness, and subjected to sterilization with ethylene oxide before packaging. Scanning electron microscopy images and detailed chemical/physical information of each composite are shown in Figure 1 and Table 1, respectively.

### 2.2. Animals

Male Sprague–Dawley (SD) rats (*n* = 28; weight, 306–320 g; Charles River, Tokyo, Japan) were allowed to acclimate to our facility for 7 days prior to the experiments. The University of Shimane Faculty of Medicine Institutional Animal Ethics Committee approved the experimental protocol, which was designed in strict accordance with the guidelines of the Care and Use of Laboratory Animals (Approval number: IZ 31–39).

### 2.3. Anesthetic Agents

Rats were anesthetized with a combination of 0.15 mg/kg medetomidine hydrochloride (Me; Nippon Zenyaku Kogyo, Tokyo, Japan), 2 mg/kg midazolam (Mi; Sandoz, Sandoz, Tokyo, Japan), and 2.5 mg/kg butorphanol (Bu; Meiji Seika Pharma, Tokyo, Japan). The three anesthetic drugs were mixed and diluted 1:2:2 with sterile saline (Hikari Pharmaceutical, Tokyo, Japan) and administered in a volume of 1 mL/100 g body weight as described previously [25].

### 2.4. Surgical Procedures

Following anesthesia with the Me-Mi-Bu agent in a volume of 1 mL/100 g body weight, rats were placed on a flat surface and surgery was performed under aseptic conditions. Following hair clipping and disinfection of the right mandibular skin, a sagittal incision approximately 1 cm in length was made through the full thickness of the mandible skin and muscle layers, and the bone surfaces were exposed. A critical bone defect was created in each rat using a trephine bur 4 mm in diameter. Then, the four groups were treated separately. Defects in the sham group were not filled with any material. In the other three groups, the defects were filled with cylinders of the biocomposites (β-TCP/PDLLA, PDLLA, or β-TCP) 4 mm in diameter and 2 mm in thickness.

The wounds were closed and the animals were maintained in a warm environment until recovery. The rats awoke 1–2 h after the operation and showed normal behavior and appetite. All rats were euthanized by inhalation of excess isoflurane (MSD Animal Health, Tokyo, Japan) during Postoperative Week 2 or 4. The mandible was harvested and soaked in 10% neutral buffered formalin (Kanto Chemical, Tokyo, Japan) for further analysis (see Figure 2 for the flow diagram of the surgery).

### 2.5. Micro-Computed Tomography

Micro-computed tomography (micro-CT) was performed by In-Vivo Science Inc. (Kanagawa, Japan) using a CosmoScan FX (Rigaku, Tokyo, Japan) at a voltage of 90 kV, current of 88 μA, and exposure time of 2 min. The field of view was 10.24 × 10.24 × 10.24 mm, the effective pixel size was 51.2 μm, and the image resolution was 20 μm. A micro-CT calcium hydroxyapatite (CaHA) phantom consisting of five cylindrical inserts containing CaHA, at a density 0, 50, 200, 800, and 1200 mg/cm^3^ (QRM, Moehrendorf, Germany), was scanned simultaneously with the samples.

### 2.6. Bone Mineral Density and the Bone Volume to Total Volume Ratio

The bone mineral density (BMD) was assessed using CTAn version 1.19+ (Bruker, Kontich, Belgium). As micro-CT primarily measures the absorption of X-rays, defined as the attenuation coefficient (AC) in mm^−1^, it was first necessary to determine the AC mm^−1^ before each BMD calculation [26]. We assumed that the X-ray attenuation within mineralized tissues is dominated by, and can be approximated as, the X-ray attenuation of the mineral compound CaHA, which has the formula Ca_5_(PO_4_)_3_(OH) [27]. We used two or more known mass concentrations of CaHA phantoms to calculate the AC mm^−1^. Relating the CaHA concentration and AC in the micro-CT images allowed us to perform calibration between these two quantities, which finally allowed us to infer BMD from the micro-CT-measured AC of mineralized bone tissue [28].

The phantoms used in this study consisted of five cylinders and were scanned simultaneously with the samples. The computed tomography (CT) images of 1200 mg/cm^3^ and 800 mg/cm^3^ cylinders were used for AC calibration. After calibration against AC, the BMD could be calculated using the follow formula:$$BMD\;\left(g/{\mathrm{cm}}^{3}\right)=\frac{AC-0.08999}{0.15}$$

After adjustment of the formula, the Digital Imaging and Communications in Medicine (DICOM) images of the rat mandibles were also imported into CTAn software. A standardized circular region of interest (ROI), consisting of over 50 slices 4 mm in diameter in the critical defect region of the right mandible, was determined and applied in all samples, and the BMD for the ROI was obtained.

The bone volume to total volume (BV/TV) ratio was calculated using the open source plugin Transform and BoneJ in Fiji software (ImageJ; NIH, Bethesda, MD, USA). The Transform plugin was used to adjust the angle of the CT images to ensure that the outline of the defect area had a perfect ring shape. Similar to the BMD calculation, the ROIs were also 4.0 mm diameter, and were focused on the mandibular critical defect regions. The ROIs were duplicated into a new stack and binarized before volume fraction analysis. The binarization process was performed under the default settings (black object on white background). BoneJ can then recognize the binarized ROIs and calculate the area/volume fraction automatically. The BV/TV ratio is output in a new log window.

### 2.7. Histochemical Staining

The samples were fixed in 10% formalin after micro-CT and then sent to Sept. Sapie Co., Ltd. (Tokyo, Japan) for histochemical staining. The mandibular samples were cut into 5-μm-thick sections from anterior to posterior. Three sections from each series were stained with hematoxylin and eosin (H&E) for histological analysis as references for the following staining procedures. Three other sections were used for immunohistochemical (IHC) staining to analyze the expression of Runx2, osteocalcin (OCN), and leptin receptor (LepR).

All sections were deparaffinized in xylene, rehydrated by passage through a graded ethanol series, and transferred into water for 5 min. The specimens were then subjected to heat treatment for 8 mins at 80 °C in sodium citrate buffer (pH 6.0) for antigen retrieval. The sections were then allowed to cool to room temperature and washed in phosphate-buffered saline (PBS, pH 7.4). Nonspecific staining was blocked by incubating the sections in 5% bovine serum albumin (BSA) in PBS for 1 h. Endogenous peroxidase activity was quenched by incubating the sections in a solution of 3% H_2_O_2_ in 10% methanol and PBS for 5 min. After washing with PBS, the sections were incubated with the following primary antibodies at room temperature for 50 min: anti-Runx2 rabbit polyclonal antibody (ab23981, 1:1000 dilution; Abcam, Tokyo, Japan); anti-OCN mouse monoclonal antibody (ab13420, 1:750 dilution; OCG3, Abcam); anti-LepR rabbit polyclonal antibody (20966-1-AP, 1:200 dilution; Proteintech, Rosemont, IL, USA). One sample in each group was treated with PBS instead of the primary antibody, to serve as the control and rule out non-specific binding of the secondary antibody. After incubation with primary antibodies, all sections were washed three times with PBS for 5 min each time, and incubated with the respective secondary antibodies at room temperature for 30 min (Runx2: No. 414191, Histofine Simple Stain Rat MAX PO (MULTI); OCN: No. 414171, Histofine Simple Stain Rat MAX PO (M); LepR: No. 101091, Histofine Simple Stain Rat MAX PO (MULTI); Nichirei Biosciences, Tokyo, Japan). The sections were then washed with PBS and incubated with diaminobenzidine substrate kit (Dako, Tokyo, Japan) for 5 min and counterstained with Mayer’s hematoxylin (Muto Pure Chemical, Tokyo, Japan) for 2 min, which yielded a brown-colored precipitate at antigen–antibody binding sites. The reaction was stopped in running tap water. Finally, all sections were dehydrated, and one drop of aqueous mounting medium was applied; the sections were then coverslipped.

### 2.8. Optical Density Assessment

All histochemical specimens were examined using an Olympus BX43F microscope (Olympus, Tokyo, Japan) with an Olympus D21-CB photo digital system (Olympus). The optical density (OD) was used to assay the intensity of Runx2, OCN, and LepR staining as described previously [29], using the open source plugin IHC profiler in Fiji software (ImageJ 2nd generation; NIH; as also described in detail elsewhere [14]). Briefly, in accordance with the color deconvolution algorithm of Ruifrok et al. [30], IHC profiler can automatically calculate the OD of hematoxylin and DAB, generating a semiquantitative score. Jafari et al. [29] suggested using the following formula to convert the outcome into a form allowing quantitative analysis:$$\begin{array}{ll} \mathrm{IHC OD score} & =\;\mathrm{percentage contribution of high positive}\times 4\\ & +\;\mathrm{percentage contribution of positive}\times 3\\ & +\;\mathrm{percentage contribution of low positive}\times 2\\ & +\;\mathrm{percentage contribution of negative}\times 1 \end{array}$$

This formula was used to calculate the OD for IHC. Under ×20 magnification, ROIs in each IHC-stained section were drawn on the newly formed tissues in the pores and areas surrounding the biomaterials in the defect region. A quantitative OD score was then calculated for each individual ROI, and the average was taken to represent the expression of each protein in different slices.

### 2.9. Statistical Analysis

The differences in BMD and BV/TV were compared using the Mann–Whitney U test. The results of IHC staining for Runx2, OCN, and LepR at 2 and 4 weeks were compared using the independent-samples Kruskal–Wallis test and the pairwise comparisons were used as the post hoc test. The significance was adjusted using the Bonferroni method; *p* < 0.05 was taken to indicate statistical significance. Statistical analyses were performed using SPSS software (ver. 27.0; IBM Corp., Armonk, NY, USA).

## 3. Results

In total, 28 rats underwent successful surgery and remained healthy until euthanasia; 4 of them were used for demonstration purposes (as shown in Figure 2), and the remaining 24 were used in the micro-CT and staining analysis.

### 3.1. Micro-CT

Micro-CT was performed at 2 and 4 weeks after surgery to analyze bone formation in the mandibular defect rat model. At 2 weeks, none of the groups showed conspicuous bone formation on micro-CT. Osteogenesis increased in all four groups from 2 to 4 weeks (Figure 3). At 4 weeks, new bone could be seen clearly surrounding the biomaterials in the β-TCP/PDLLA, β-TCP, and PDLLA groups (Figure 3E–G; Figure 3M–O). Large amounts of new bone grew on the buccal and lingual sides of the material, especially in the β-TCP/PDLLA and the β-TCP group.

At 2 and 4 weeks, the BMD and BV/TV values differed between the β-TCP/PDLLA and β-TCP groups. In addition, the BMD and BV/TV values were higher in the β-TCP group than in the β-TCP/PDLLA group at 2 and 4 weeks, and the β-TCP group showed more active bone regeneration than the β-TCP/PDLLA group. However, the difference was not significant. (Figure 4).

### 3.2. Histochemical Staining

#### 3.2.1. H&E Staining

Hematoxylin and eosin (H&E) staining was used for histochemical reference. The H&E-stained sections showed differences in the extent of bone regeneration among the β-TCP/PDLLA, β-TCP, and PDLLA groups. In the β-TCP/PDLLA group, multinucleate cells were observed around the scattered materials (Figure 5E,M); this was not seen in the β-TCP group. The degradation of PDLLA was slower than that of the other two materials, and the outline of the material was very clear, especially in the photographs taken at 2 weeks. Even 4 weeks after implantation, the material was still relatively intact compared to β-TCP/PDLLA and β-TCP (Figure 5K). Bone regeneration was not obvious in the sham group.

#### 3.2.2. IHC Staining

In all three groups, Runx2 accumulated in newly formed bone and tissue adjacent to the periosteum, at both 2 and 4 weeks (Figure 6). At 2 weeks, the amount of newly formed bone within the biomaterials was limited, and Runx2 was predominantly expressed in the peripheral areas outside of each biomaterial. The differences between the β-TCP/PDLLA group and the β-TCP group were statistically significant (*p* < 0.05, Figure 6). At 4 weeks, due to the degradation of biomaterials and an increase in the amount of newly formed bone, Runx2 expression inside the materials could be detected more easily. The intensity of Runx2 signals was markedly higher in the β-TCP and β-TCP/PDLLA groups than in the PDLLA group (*p* < 0.05, Figure 6), but the difference between the β-TCP and β-TCP/PDLLA groups was not significant (*p* > 0.05, Figure 6) at 4 weeks.

Positive staining for OCN revealed functional osteoblast formation and osteoid matrix deposition (Figure 7). At Week 2, new bone was clearly seen inside the β-TCP and positive staining for OCN was widespread in the bone matrix. However, in the β-TCP/PDLLA and PDLLA groups, OCN was predominantly expressed around but not within the materials. Among the three biomaterials, the intensity of OCN staining was greater in the β-TCP/PDLLA group compared to the β-TCP and PDLLA groups (significance was found between the β-TCP/PDLLA and β-TCP groups, *p* < 0.05, Figure 7) at Week 2. At Week 4, more regenerated bone had infiltrated into the biomaterials. Regeneration was typically initiated in the area between the host bone and materials. However, in the β-TCP group, we also detected newly formed bone in the center of the biomaterial, which was not seen in the other two groups (Figure 7B,F). The differences in OCN staining intensity among the three groups were not significant at Week 4 (*p* > 0.05, Figure 7).

LepR expression was detected surrounding the newly formed bone or close to the periosteum at 2 and 4 weeks. At 2 weeks, a cell monolayer positive for LepR expression was clearly seen around the regenerated bone (Figure 8E’–G’; Figure 8M’–P’). These observations were most obvious in the β-TCP group, followed by the β-TCP/PDLLA and PDLLA groups. However, the level of LepR expression was highest in the PDLLA group (*p* < 0.05, Figure 8), consistent with the observations regarding Runx2 expression. At 4 weeks, the β-TCP/PDLLA groups showed the highest level of LepR expression, and LepR-positive cells adhered to the edges of the biomaterials. This accumulation pattern was not as conspicuous in either the β-TCP or PDLLA group. In the β-TCP group, LepR-positive cells were also seen adjacent to the area of newly formed bone, while in the PDLLA group the LepR-positive cells accumulated mainly on the outside part of the biomaterials. Levels of LepR expression were significantly higher in the β-TCP/PDLLA group than in the β-TCP and PDLLA groups (*p* < 0.05, Figure 8), but the difference between the β-TCP and PDLLA groups was not statistically significant (*p* > 0.05, Figure 8).

## 4. Discussion

The foundations of tissue engineering (TE) were described in 1993 by Langer and Vacanti [31]. The goal of TE is to provide a route to tissue repair and regeneration. TE was developed as an alternative approach to allow replacement of injured and lost tissues, to address the significant shortage of donors [32] and difficulties in overcoming host immune responses that lead to graft failure [33].

Bone is a highly dynamic tissue with diverse roles in human physiology and is also among the tissues that most frequently requires replacement due to loss from osteoporosis, trauma, or cancer resection [34]. Repairing large bone defects is a significant challenge, particularly in the maxillofacial area due to the complex structures, load bearing, and aesthetic considerations [35]. Autogenous bone grafting is still deemed the gold standard but has various limitations, such as donor site morbidity, lack of availability, etc. [36]. Accordingly, several bone graft substitutes, consisting of biocompatible and biodegradable biomaterials, have been introduced over the last 25 years [37]. The novel nano-biocomposite β-TCP/PDLLA, developed by Teijin, consists of 70 wt% β-TCP and 30 wt% PDLLA; it has a porosity of 70% ± 5%, which is similar to human cancellous bone. This was the first in vivo study to test the biocompatibility and bone regeneration-promoting capability of β-TCP/PDLLA.

In our previous studies [14,15,16,17], we found that rapid bone regeneration was limited to the first 4 weeks postoperatively. Then, the growth rate slowed markedly from 4 to 8, and even to 16 weeks, as was also demonstrated in other studies [38]. In this study, therefore, we investigated the initial response and performance of this nano-biocomposite in the early stages of bone healing at 2 and 4 weeks.

Micro-CT was used to investigate bone regeneration in all four groups. However, as PDLLA is not radiopaque, comparisons were only carried out in the β-TCP/PDLLA and PDLLA groups. The β-TCP composite has higher porosity and a smaller pore size than β-TCP/PDLLA in the original state, which also caused differences in micro-CT images. At both 2 and 4 weeks, the β-TCP group showed higher BMD and BV/TV values. As mentioned above, the properties of the composite itself can influence the radiological outcome, and β-TCP was indeed shown to have superior capacity for bone integration [18]. Although a difference in BMD and BV/TV between these two groups was evident, the statistical analysis did not show significance. We assume this was due mainly to the limited sample size. Newly formed bone was detected in all three groups at a very early stage, i.e., at 2 weeks, and the new bone extended from the host mandibular bone tissue and fused with β-TCP. It was difficult to discern the immature bone tissue and β-TCP in some parts of the CT images. Particularly at 4 weeks, the degree of fusion between host bone and β-TCP was increased, and new bone was also found in the pores of the material. However, the bone formation was distinct in β-TCP/PDLLA, fusion between bone and the material was rare, and most newly formed bone was only detected inside the pores.

The IHC results revealed more details about osteoblast differentiation and maturation. This was the first study in the field of biomaterials research to use LepR for detection of the differentiation of precursor cells during bone regeneration. LepR is a type I cytokine receptor [39], which is encoded in humans by the LepR gene [40,41]. Previous studies provided in vivo evidence that LepR^+^ cells differentiate not only into osteoblasts, but also into adipocytes during tissue regeneration after injury [42]. Mengyu et al. [43] reported that LepR^+^Runx2-GFP^low^ cells showed increased levels of Runx2, Osterix, and type I collagen α expression and form multilayered structures close to the bone surface, resulting in the generation of mature osteoblasts. This study also included analysis of Runx2 and OCN expression to evaluate osteoblast differentiation. Runx2 and LepR showed very similar expression trends and identical distributions in all three groups at 2 and 4 weeks, with positive cells detected predominantly adjacent to the newly formed bone. Both Runx2 and LepR are biomarkers of these progenitor cells with stemness, which have the potential to differentiate into osteoblasts or adipocytes [44]. The PDLLA group showed the highest levels of Runx2 and LepR expression at 2 weeks, but the levels decreased at 4 weeks. With the degradation of β-TCP/PDLLA, the pore size increased and more cells infiltrated into these areas; the levels of Runx2 and LepR expression in the β-TCP/PDLLA group surpassed those in the PDLLA group at 4 weeks. The β-TCP group also showed the same findings, although the OD of LepR was lower than that in the PDLLA group at 4 weeks, but the difference was not significant. OCN is a major component of bone extracellular matrix and a marker of osteogenesis, and is mainly secreted by mature osteoblasts [45]. The level of OCN expression was higher in the β-TCP/PDLLA group than the other two groups at 2 weeks, but was similar among all three groups at 4 weeks, suggesting that this biomaterial may provide favorable conditions for osteoblast cell differentiation and maturation in the initial period. This was consistent with a previous study based on calcium phosphate and polymer composite [46].

The complete degradation of pure-phase β-TCP in humans and animal models occurs within a 6–12-month period [47], which is far quicker than PDLLA (12–30 months) [7,48,49] and β-TCP/PDLLA (9–19 months) [50,51]. The rapid resorption will provide more space for tissue ingrowth and release large amounts of Ca^2+^ and P^5+^ ions into the milieu, which are essential for osteoblast maturation and bone formation [52]. However, the degradation of PDLLA is very complicated and occurs mainly through ester group hydrolysis, which is divided into two stages [52]. In the first stage, the backbone of the polymer begins to break down and the molecular weight decreases markedly due to the penetration of water molecules. In the second stage, the ester groups on the molecular chain are exposed to water, and hydrolytic cleavage takes place [51]. After hydrolysis, PDLLA degrades into d-lactide and l-lactide and the carboxyl end groups will diffuse into the environment, which also causes a decrease in pH resulting in local acidity. The acidic milieu is considered unfavorable for bone regeneration, and the ideal pH value should be around 7.4–8.6 for osteoblast differentiation [53].

On the other hand, as a natural metabolite in animals, l-lactide will not be harmful to the bodies of humans or animals. It can undergo direct oxidation decomposition to CO_2_ and H_2_O in the presence of sufficient oxygen or enter into the active gluconeogenesis pathway under conditions of hypoxia [54]. However, d-lactide cannot be metabolized in the body, and may cause acidosis if present in excess [55]. The elimination of d-lactide requires phagocytosis by macrophages, which may also explain the accumulation of inflammation cells or multinucleate cells in the sections of the β-TCP/PDLLA rather than the β-TCP composite [52] (Figure 9). The accumulated multinucleate cells competitively occupied the scattered material surface with osteoblasts. Meanwhile, the released nanoparticles of calcium phosphate may also be blocked by this cellular behavior, thus causing a delay or cessation of osteoblast differentiation [52,55].

In our study, the ratio of β-TCP to PDLLA was 70 wt% to 30 wt%, and that of PDLA to PLLA was 50 mol% to 50 mol%. The polymer content in this composite was comparatively low, but bone regeneration was still not satisfactory, and the results were inconsistent with a previous study [56]. There are two possible explanations for this. First, a rodent with a small body size may not metabolize this amount of polymer in a short time, and studies in larger animals, such as rabbits, may provide different results, as in the study of Lin et al. [52]. Second, the polymer content may need to be further modified, such as from 50:50 mol% PDLA/PLLA to 35/35/30 mol% PDLA/PLLA/PDLLA, as described by Zhang et al. [57]; this wider molecular distribution may facilitate hydrolysis while maintaining the same mechanical strength of the composite [57]. Yang et al. [46] reported that 2:1 β-TCP/PLLA significantly increased OCN production by cells. In our previous study [17] and another ongoing study, we also found that HA/PLLA sheet and HA/PLLA/PGA [58] demonstrated ideal bone regeneration outcomes. This may be helpful when considering the configuration of new (fourth-generation) bioresorbable biomaterials.

In this pilot study, only 2- and 4-week time points were used to assay the bone regeneration of different biocomposites. Further prolongation of the observation period may reveal more biological features. In addition, as PDLLA is not radiopaque, no radiological information about it can be gained using the procedures outlined herein. Therefore, a modified testing procedure should be explored in future studies.

Our in vivo results indicated that β-TCP/PDLLA initially promoted progenitor cell and osteoblast differentiation; detailed knowledge of the biological and physical features of this biomaterial will be useful for subsequent trials. Although β-TCP has been widely used in clinical practice in oral and maxillofacial surgery, pure β-TCP is very fragile and brittle, and cannot be reshaped. For regular fractures or defects, β-TCP can work well, but if the defect is not regular or is located in a load-bearing site, β-TCP is not always sufficiently effective. PDLLA has similar drawbacks, as it is a pure organic material with a long degradation period that causes body reactions, so cannot be used as a load-bearing biomaterial. Therefore, it is vital to identify biomaterials that are able to overcome these drawbacks. β-TCP/PDLLA can be shaped during an operation by thermal methods, or using a scalpel; this is beneficial to surgeons during complex defect surgery. Meanwhile, it is also sufficiently solid to withstand load-bearing stress, and its degradation rate can be adjusted by changing the ratio of inorganic/organic particles [57]. These characteristics of β-TCP/PDLLA are sufficiently important to merit further improvement. We plan to further modify the chemical compounds in the biomaterials, to enhance the capacity and thus promote bone regeneration in maxillofacial bony defects for the development of promising alternative biomaterials for bone graft surgery.

## 5. Conclusions

Based on our pilot in vivo study, the novel β-TCP/PDLLA nano-biocomposite showed good biocompatibility, bioresorbability, and bioactive/osteoconductivity. In addition to these unique biological features, β-TCP/PDLLA can easily be reshaped during surgery. Moreover, its compressive strength is similar to natural human bone tissue, and its degradation rate can be regulated by adjusting the ratio of its internal particles. The mechanisms underlying how this nanomaterial affects osteoblastic cell differentiation and osteogenesis warrant further study and investigation.

## Figures and Tables

**Figure 1 nanomaterials-11-00303-f001:**
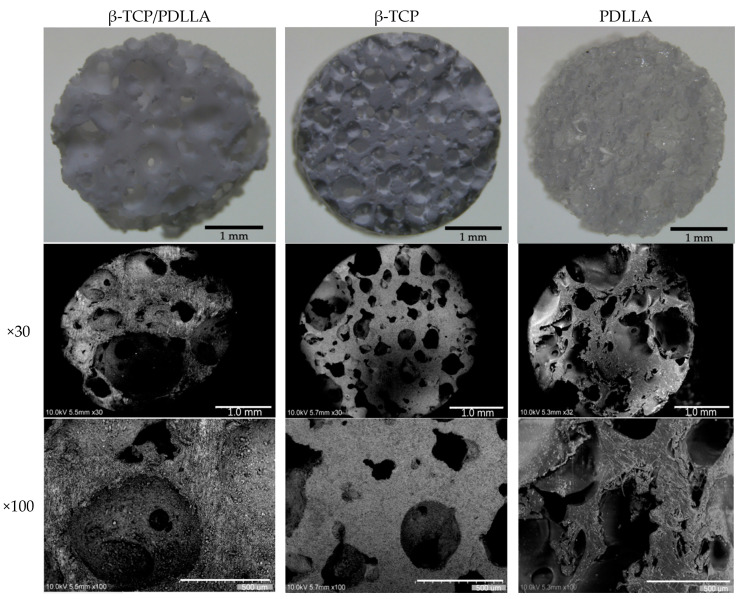
Stereomicroscopy and scanning electron microscopy images (×30 and ×100 magnification) of β-tricalcium phosphate/u-HA/poly-d/l-lactide (β-TCP/PDLLA), β-TCP, and PDLLA.

**Figure 2 nanomaterials-11-00303-f002:**
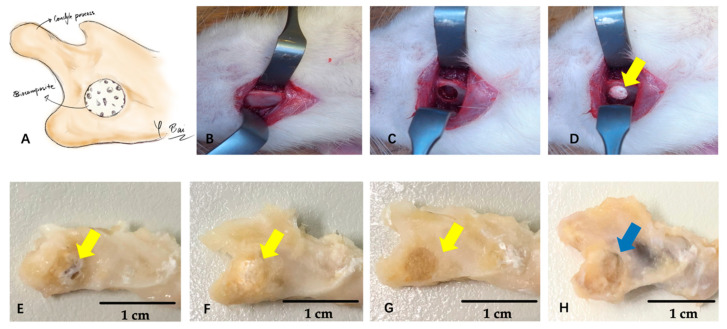
Flow diagram of the surgery and mandible specimens. (**A**) Relative position of the mandible and biocomposite. (**B**) Exposure of the right lateral mandible. (**C**) Creation of a critical defect. (**D**) Filling with different biocomposites. (**E**) Mandible with β-TCP/PDLLA. (**F**) Mandible with β-TCP. (**G**) Mandible with PDLLA. (H) Mandible with defect only. (**E**–**H**) All photographs depict the 2-week postoperative group. Yellow arrow indicates the biocomposite. Blue arrow indicates the defect. Scale bar: 1 cm. The materials in E–H were only used for demonstration purposes, and were not included in the micro-computed tomography (micro-CT) or staining experiments.

**Figure 3 nanomaterials-11-00303-f003:**
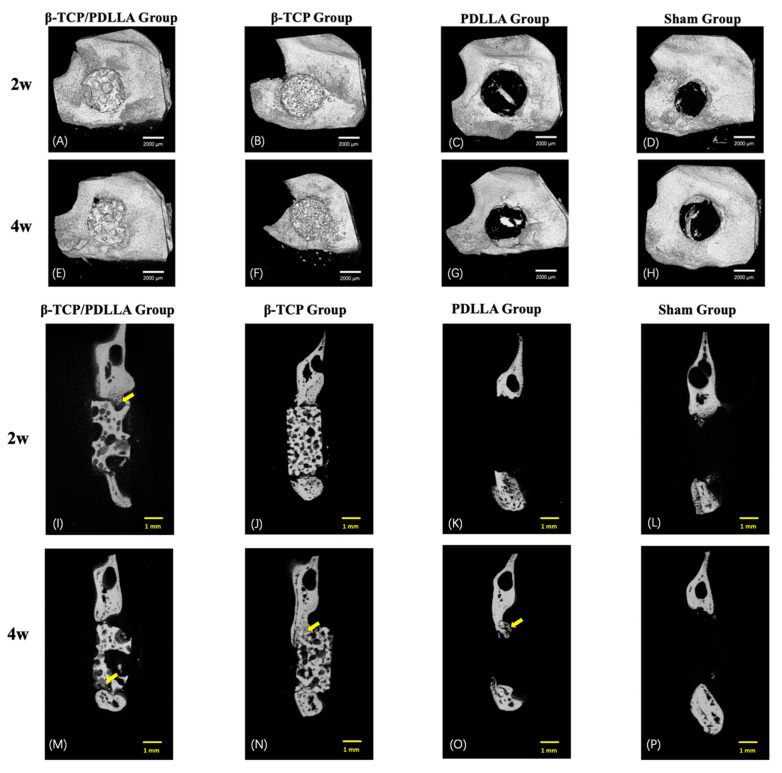
Micro-CT images of the mandible defects from the sagittal and coronal plane sections at 2 weeks (**A**–**D**,**I**–**L**; *n* = 3) and 4 weeks (**E**–**H**,**M**–**P**; *n* = 3). (**A**,**E**,**I**,**M**) Group 1: β-TCP/PDLLA group. (**B**,**F**,**J**,**N**) Group 2: β-TCP group. (**C**,**G**,**K**,**O**) Group 3: PDLLA group. (**D**,**H**,**L**,**P**) Group 4: sham group. The yellow arrows indicate the newly formed bone. Scale bars: 2000 µm (white) and 1 mm (yellow).

**Figure 4 nanomaterials-11-00303-f004:**
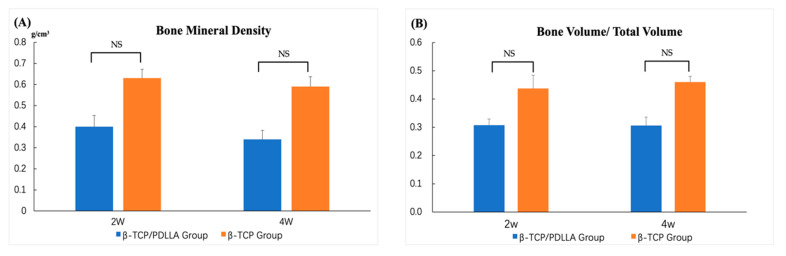
Micro-CT analysis of the (**A**) bone mineral density (BMD) and (**B**) bone volume/total volume (BV/TV) of the β-TCP/PDLLA and β-TCP groups. Mann–Whitney U Test; NS, not significant. Error bars indicate standard deviation.

**Figure 5 nanomaterials-11-00303-f005:**
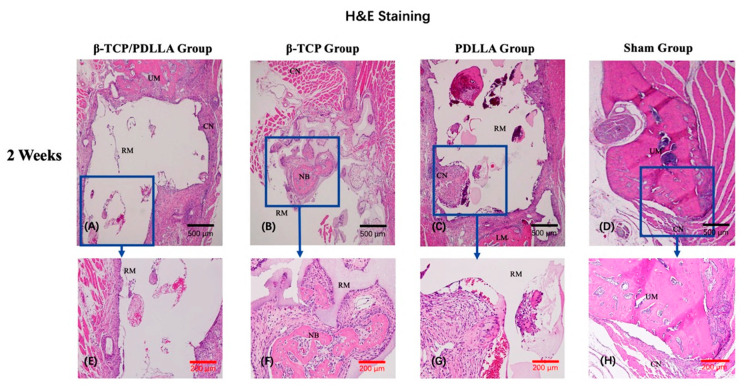
Hematoxylin and eosin staining of specimens in the β-TCP/PDLLA (*n* = 3), β-TCP (*n* = 3), PDLLA (*n* = 3), and sham (*n* = 3) groups. (**A**–**H**,**E’**,**E’’**) Sections at 2 weeks after mandibular defect surgery. (**I**–**P**,**M’**,**M’’**) Sections at 4 weeks after mandibular defect surgery. UM, upper mandible; LM, lower mandible; RM, residual material; NB, newly formed bone; CN, connective tissue. All slices are shown at ×4, ×10 and ×40 magnification. Scale bars: 500 μm (black), 200 μm (red), and 50 μm (blue). The black arrows indicate the multinucleate cells observed in the H&E slices, adjacent to the scattered biocomposites.

**Figure 6 nanomaterials-11-00303-f006:**
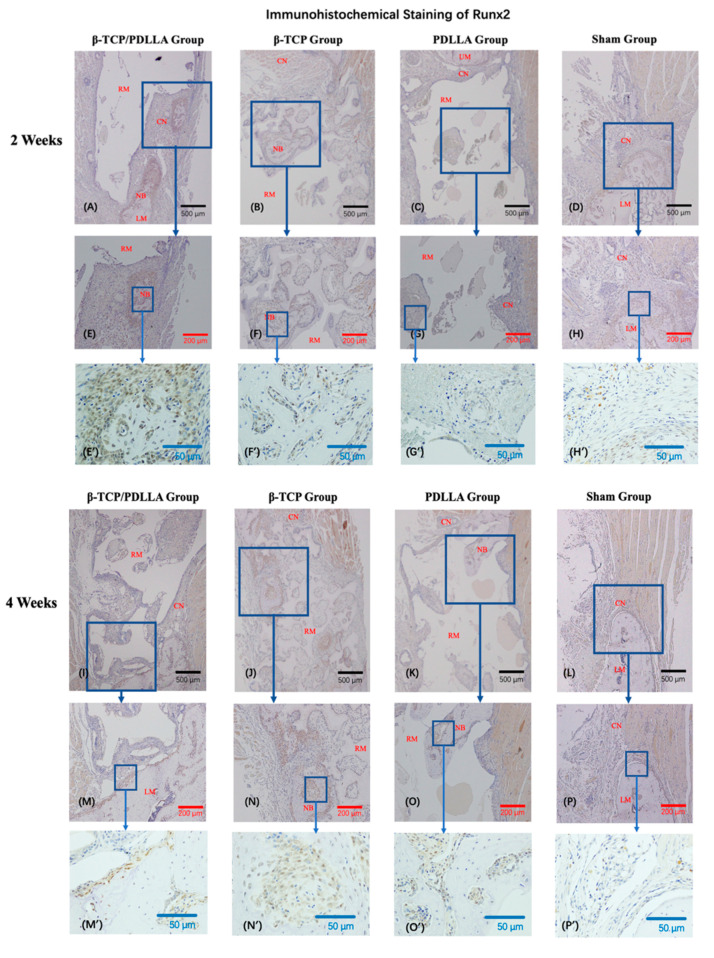
Immunohistochemical staining of Runx2 in the β-TCP/PDLLA (*n* = 3), β-TCP (*n* = 3), PDLLA (*n* = 3), and sham (*n* = 3) groups. (**A**–**H**,**E’**–**H’**) Sections at 2 weeks after mandibular defect surgery. (**I**–**P**,**M’**–**P’**) Sections at 4 weeks after mandibular defect surgery. UM, upper mandible; LM, lower mandible; RM, residual material; NB, newly formed bone; CN, connective tissue; OD, optical density. All slices are shown at ×4, ×10 and ×40 magnification. Scale bars: 500 μm (black), 200 μm (red), and 50 μm (blue). (**Q**) OD of Runx2 was analyzed using the Kruskal–Wallis test and the pairwise comparisons were used as the post hoc test; * *p* < 0.05; NS, not significant. Error bars indicate standard deviation.

**Figure 7 nanomaterials-11-00303-f007:**
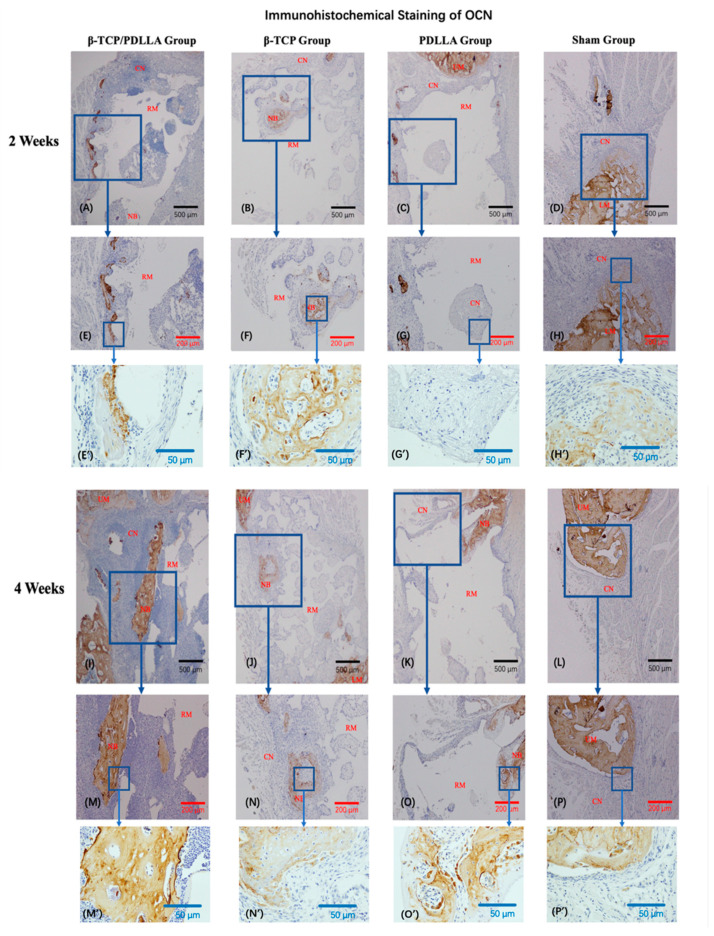
Immunohistochemical staining of OCN in the β-TCP/PDLLA (*n* = 3), β-TCP (*n* = 3), PDLLA (*n* = 3), and sham (*n* = 3) groups. (**A**–**H**,**E’**–**H’**) sections at 2 weeks after mandibular defect surgery. **(I**–**P**,**M’**–**P’**) Sections at 4 weeks after mandibular defect surgery. UM, upper mandible; LM, lower mandible; RM, residual material; NB, newly formed bone; CN, connective tissue; OD, optical density. All slices are shown at ×4, ×10 and ×40 magnification. Scale bars: 500 μm (black), 200 μm (red), and 50 μm (blue). (**Q**) The OD of OCN was analyzed using the Kruskal–Wallis test and the pairwise comparisons were used as the post hoc test; * *p* < 0.05; NS, not significant. Error bars indicate standard deviation.

**Figure 8 nanomaterials-11-00303-f008:**
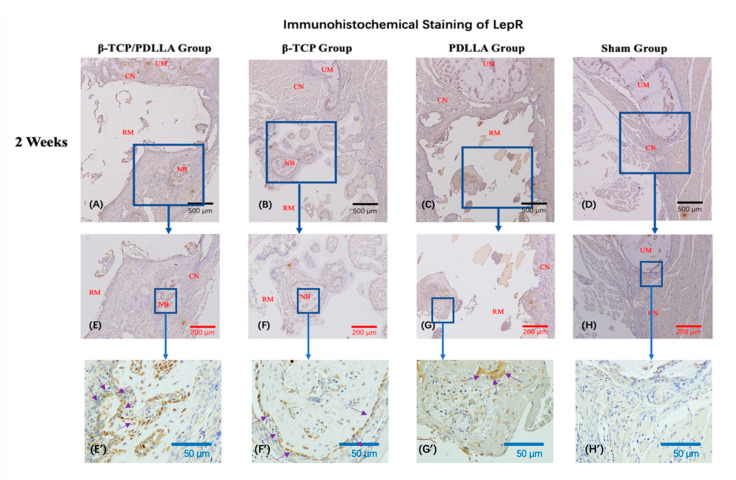
Immunohistochemical staining of LepR in the β-TCP/PDLLA (*n* = 3), β-TCP (*n* = 3), PDLLA (*n* = 3), and sham (*n* = 3) groups. (**A**–**H**,**E’**–**H’**) Sections at 2 weeks after mandibular defect surgery. (**I**–**P**,**M’**–**P’**) Sections at 4 weeks after mandibular defect surgery. UM, upper mandible; LM, lower mandible; RM, residual material; NB, newly formed bone; CN, connective tissue; OD, optical density. All slices are shown at ×4, ×10 and ×40 magnification. Scale bars: 500 μm (black), 200 μm (red), and 50 μm (blue). The purple arrows indicate the monolayer positive for LepR expression in each group. (**Q**) The OD of LepR was analyzed using the Kruskal–Wallis test and the pairwise comparisons were used as the post hoc test; * *p* < 0.05; NS, not significant. Error bars indicate standard deviation.

**Figure 9 nanomaterials-11-00303-f009:**
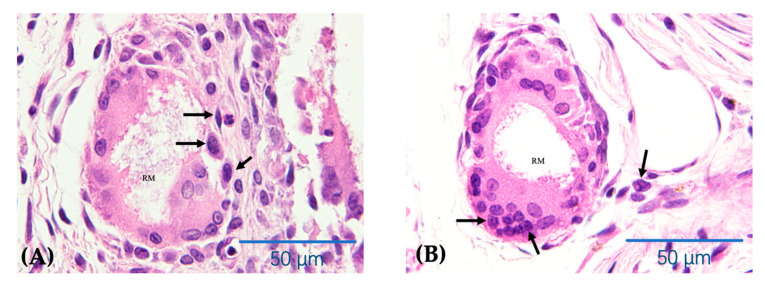
Multinucleate cells accumulated around the scattered β-TCP/PDLLA material at 2 weeks (**A**) and 4 weeks (**B**). Scale bars: 50 μm. Black arrows indicate the multinucleate cells surrounding the scattered biocomposite. RM: residual material. (**C**) Overview of osteoblast matsuration and new bone formation. The stem cells predominantly derived from bone marrow and periosteum undergo a condensation process and form primitive progenitor cells. With the degradation of porous biocomposites, such as β-TCP/PDLLA, large amounts of Ca^2+^ and P^5+^ ions are released into the milieu, which is essential for further differentiation of progenitor cells into immature osteoblasts [52]. Finally, immature osteoblasts differentiate into mature and functional cuboidal cells that secrete bone matrix and form new bone. (**D**) The accumulated multinucleate cells may competitively occupy the scattered material surface with osteoblasts. Meanwhile, the released Ca^2+^ and PO_4_^3−^ ions may also be blocked by this behavior, causing a delay or cessation of osteoblast differentiation [52,55].

**Table 1 nanomaterials-11-00303-t001:** Chemical and physical information of the biomaterials.

	Components (wt%)		Size (mm) Av. ± SD		Porosity (%) Av. ± S.D.	Weight (g) Av. ± SD	Density (g/cm^3^) Av. ± SD
	β-TCP	PDLLA	Diameter	Thickness			
β-TCP/PDLLA	70	30	4.0 ± 0.2	2.0 ± 0.2	70 ± 5	0.017 ± 0.001	0.68 ± 0.04
β-TCP	100	0	4.0 ± 0.2	2.0 ± 0.2	77.5 ± 4.5	0.016 ± 0.001	0.69 ± 0.06
PDLLA	0	100	4.0 ± 0.2	2.0 ± 0.2	70 ± 5	0.008 ± 0.001	0.37 ± 0.03

## Data Availability

All data have been illustrated in the manuscript.

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
