# Peer review of "Feasibility of Application of the Newly Developed Nano-Biomaterial, β-TCP/PDLLA, in Maxillofacial Reconstructive Surgery: A Pilot Rat Study"

_nanomaterials, 2021, doi:10.3390/nano11020303_

Round 1

Reviewer 1 Report

The manuscript looks better. 

Author Response

Dear reviewer 1

We are truly grateful for your recognition of our article. First of all, we do thank your kind commends.

Based on other reviewers’ comments and suggestions, we have made careful modifications into our original manuscript. All changes made to the main text are in red. We want to say thank you for your appreciation very much.

Sincerely yours,

The corresponding author

Prof. Takahiro Kanno

Reviewer 2 Report

The paper describes the in vivo evaluation of a composite material, beta-TCP/PDLLA (70:30 wt%). Provided that D-Lactide is not really appropriate as a biomaterial due to degradability issues (something that is actually acknowledged by the authors in the discussion - p. 374), it is not clear why authors decide to use PDLLA instead of just PLLA.

To evaluate the material, a defect was practiced in rat mandibles and either left empty (control), or filled with beta-TCP/PDLLA. PDLLA and beta-TCP alone were also used as controls. Three animals were used for each experimental condition. Results were analyzed by micro-CT, and Histology, including Hematoxylin-Eosin histochemical staining and immunohistochemical detection of Runx2, osteocalcin and LepR.

After reading the abstract, it is expected that the new composite material (beta-TCP and PDLLA) will bear some advantage over already used materials, or at least over the composite elements on its own. However, results show a low performance of the composite, and even authors admit in the discussion. Then, what’s relevant about this novel material?

There are also some concerns about the experimental design:

The number of animals used for each experimental situation is small, and the use of microCT is arguable, given that

  • PDLLA is not detectable by this method, as the own authors concede in the Discussion (p. 396)
  • It is hard to distinguish beta-TCP from bone, as the authors admit in p. 320.

The use of histochemical and immunohistochemical techniques is also questionable. They have not used any of the staining recommended for bone tissue, like von Kossa, which would have allowed them a quantification of new bone tissue in the defect. Even if they lack the technology to obtain undecalcified tissue sections, they could have used Masson-Goldner’s trichrome or picrosirius staining. They have immunostained Runx2 and LepR to label osteoprogenitor cells, and osteocalcin to label osseous extracellular matrix, which is correct, but the images shown in the manuscript are not of enough quality to judge is even the staining is specific. Besides, without a proper study and quantification of the extracellular matrix formed in the defect, the data offered by Runx2, LepR and OCN immunostaining are partial.

Major Comments:

Methods, p. 86. “The porosity calculated from the apparent density…” This calculation should be explained in detail.

Methods-Surgical procedure. Photographs should be taken during the surgery to illustrate the operation field. Otherwise, it is difficult to understand the location of the defect and its size in relation to the whole mandible size.

Methods-Micro-CT analysis. Besides the use of phantoms, BMD of the normal bone (e.g. bone in a similar location in the non-operated left mandible), should have been obtained and used as a reference.

Methods-Immunohistochemistry. No control without primary antibody incubation was performed to assess that the observed staining patterns were specific.

Methods-Immunohistochemical signal quantification by optical density assessment. How many sections of each immunostaining were used in the optical density analysis? Where they from different parts of the defect or from a particular one? (i.e. did they cover all the defect or did they come all from its centre?)

Results-Micro-CT analysis, p. 208. The authors claim that “new bone could be seen clearly surrounding the biomaterials”. From the images provided in figure 2, it is hard to see new bone clearly, because the biomaterial and the bone are not distinguishable. Provided the authors used phantoms to dissect out the attenuation coefficient of bone and beta-TCP, images offered in fig. 2 should be segmented, so that bone and biomaterial could be clearly differentiated.

Results-Histology. In general, the images provided are small, dark and have low resolution. It is appreciated that figures provide, for each biomaterial tested, a wide-field image showing the defect, and then a higher magnification of a highlighted area within the wide-field. The formers are good to let the reader know how the defect looks like at each time point. But the latter should provide enough magnification to observe what authors claim is observable within them. And this is not the case. Higher-magnification images accompanying the wide-field ones are not high-mag. enough. For example, Runx2 is a transcription factor, and LepR is a membrane receptor. Then, Runx2 and LepR IHQ images should be zoomed enough to observe the label at the cell level.

Authors tend to describe elements as clearly visible in the figures, without pointing out at them with arrows, asterisks or some other type of indicators. Examples:

- p. 225-226: “multinucleated cells or macrophages were observed around the scattered materials (Figure 4, E and M)”. Honestly, where are the said macrophages?

-p. 265: “a cell monolayer positive for LepR expression was clearly seen around the regenerated bone (Figure 7, A–C)”. Said monolayer is not visible en figures 7A and 7C. It is hardly intuitable in figure 7E, but how does the reader know he/she is looking at the same spot the author has in mind?

Authors also describe elements that are not actually discernible with the given technique. For example: going back to the macrophages described in pages 225-226. Macrophages are not visible with H-E staining. They should be highlighted with a proper immunostaining.

By the way, macrophages are not multinucleate cells. Maybe authors actually mean osteoclasts. Osteoclasts are visible to a well-trained eye with H-E, but the right way to show them is a TRAP staining.

Minor comments:

Abstract, pp. 22-23. LepR is defined as "a biomarker for precursor cell with stemness". Redundant, any precursor cell, by definition, retains some stemness.

Results, p 254. “Positive staining for OCN revealed functional osteoblast formation”. Osteocalcin is an extracellular-matrix protein. Positive staining for OCN would rather reveal osteoid matrix deposition.

Author Response

Dear reviewer 2

We are truly grateful for your critical comments and thoughtful suggestions on our manuscript. Based on all of these comments and suggestions, we have made careful modifications into our original manuscript. All changes made to the main text are in red. We want to say thank you for your helpful suggestions, which were very supportive of further improving this manuscript. You will find our point-by-point responses to your comments/ questions as below.

Sincerely yours,

The corresponding author

Prof. Takahiro Kanno

Reviewer 3 Report

Point 01

With 28 rats, one sample per rat, and 4 groups (sham plus the 3 groups of biocomposite), I can assume that there were 7 rats per group. If the rats were euthanized during postoperative week 2 or 4, how many rats per group were euthanized at postoperative week 2 and how many rats per group were euthanized at postoperative week 4? I could read that each group was allocated 3 samples depending on the test. This is not entirely clear and the authors need to better explain this.

Point 02

Provide the confidence intervals for each group for every comparison.

Point 03

With only 7 samples per group, the authors have chosen the wrong statistical test to compare the results. They should have chosen non-parametric tests instead of Student’s t-test and ANOVA.

Yet, I could read that each group was allocated 3 samples depending on the test, meaning that the sample size is too small to invoke any meaningful statistical test results. Where are the results of a power analysis? When sample sizes are small, and a power analysis is not available, one may fail to reject the null hypothesis then the true state of nature is very different from what is stated in the null hypothesis.

Point 04

“Further prolongation of the observation period may reveal more biological features.”

Why? What else could possibly happen after that? And how much faster is the bone metabolism of rats compared to humans?

Point 05

“Also, as PDLLA is radiopaque, no radiological information about it can be gained using the procedures outlined herein. Therefore, a modified testing procedure should be explored in future studies.”

Could the authors cite an example of a modified testing procedure for that?

Point 06

“The novel nano-biocomposite, β-TCP/PDLLA, showed good biocompatibility and osteoconductivity”

“Good” is relative. Express this better.

Besides, it was said that this is a “promising” biomaterial, but several other biomaterials also show “good biocompatibility and osteoconductivity”. Then, how “promising” for these biocomposites used here compare with “promising” for other biomaterials? This was not discussed.

Point 07

“Although the amount of regenerated bone in the β-TCP/PDLLA group was not surpassing too much compared with the other two groups (…)”

What does “was not surpassing too much” mean? This is totally unclear. Even due to the limited number of samples in each group. The authors are facing a big issue here, with only 3 samples per group in the comparisons between groups.

Author Response

Dear reviewer 3

We are truly grateful for your critical comments and thoughtful suggestions on our manuscript. Based on these comments and suggestions, we have made careful modifications into our original manuscript. All changes made to the main text are in red. We want to say thank you for your helpful suggestions, which were very supportive of further improving this manuscript. You will find our point-by-point responses to your comments/ questions as below.

Sincerely yours,

The corresponding author

Prof. Takahiro Kanno

Round 2

Reviewer 3 Report

The authors have improved the manuscript on most of the points.

However, the main issue remains. On my previous point 03, the authors replied:

“On the theory, whether a non-parametric or a parametric test was be adopted in a study predominantly based on whether the data belong to the normal distribution or not, which can be calculated by Kolmogorov-Smironov or Shapiro-Wilk test. We verified the distribution of all the value in each group, most of them belonged to the normal distribution. On the other hand, because of the Central Limit Theorem(in many situations, when independent random variables are added, their properly normalized sum tends toward a normal distribution even if the original variables themselves are not normally distributed), most independent variables belong to the normal distribution, or can convert into normal distribution by several methods. To be honest, how to choose different methods for analysis in reality work is still be in the controversial, due to the non-parameter test with less efficient as compared to parametric test, we think in this case student-t and ANOVA will better for current study.”

Incorrect. It is impossible to properly verify normality in groups with only 3 samples. When you have a really small sample, you might not even be able to ascertain the distribution of your data because the distribution tests will lack sufficient power to provide meaningful results.

“Can convert into normal distribution by several methods”. Which methods? Provide references.

“how to choose different methods for analysis in reality work is still be in the controversial, due to the non-parameter test with less efficient as compared to parametric test”: It is controversial for a higher amount of samples in each group, not when groups have only 3 samples.

“the non-parameter test with less efficient as compared to parametric test”: This is not an excuse to run away from the most adequate tests in these cases (very small number of samples in each group), which are non-parametric tests, as (already mentioned above) normality cannot be properly verified. That is why I wrote in the my previous review that the authors are facing a big issue here, as nonparametric tests have less power to begin with and it’s a double whammy when you add a small sample size on top of that. But you cannot run away from that when there are only 3 samples in each group. Then, I will give one more chance to the authors to address this issue.

Author Response

Dear reviewer 3

We are truly grateful for your critical comments on our manuscript. Based on your comments and suggestions, we have made careful modifications about the statistical analysis in our original manuscript. All changes made to the main text are in red. We want to say thank you for your helpful suggestions once again, which were very supportive of further improving this manuscript. You will find our point-by-point responses to your comments/ questions as below.

Sincerely yours,

Prof. Takahiro Kanno

Comments and Suggestions for Authors

The authors have improved the manuscript on most of the points.

However, the main issue remains. On my previous point 03, the authors replied:

“On the theory, whether a non-parametric or a parametric test was be adopted in a study predominantly based on whether the data belong to the normal distribution or not, which can be calculated by Kolmogorov-Smironov or Shapiro-Wilk test. We verified the distribution of all the value in each group, most of them belonged to the normal distribution. On the other hand, because of the Central Limit Theorem(in many situations, when independent random variables are added, their properly normalized sum tends toward a normal distribution even if the original variables themselves are not normally distributed), most independent variables belong to the normal distribution, or can convert into normal distribution by several methods. To be honest, how to choose different methods for analysis in reality work is still be in the controversial, due to the non-parameter test with less efficient as compared to parametric test, we think in this case student-t and ANOVA will better for current study.”

Incorrect. It is impossible to properly verify normality in groups with only 3 samples. When you have a really small sample, you might not even be able to ascertain the distribution of your data because the distribution tests will lack sufficient power to provide meaningful results.

 Response: Thanks for your critical and insightful suggestions. We have redone the statistical analysis according to your opinions by utilizing the nonparametric analysis for all comparisons.

“Can convert into normal distribution by several methods”. Which methods? Provide references.

Response: Thanks for your question. Transform data to better fit the normal distribution is usually done using mathematical functions such as square root or logarithm [1-3]. We do hope these references below might have answered your question. We all revised the statistical analysis according to your suggestions.

  1. Bland, J. Martin, and Douglas G. Altman. "Statistics notes: Transforming data." Bmj 312.7033 (1996): 770.
  2. Krithikadatta, Jogikalmat. "Normal distribution." Journal of conservative dentistry: JCD 17.1 (2014): 96.
  3. Bland, J. Martin, and Douglas G. Altman. "Transformations, means, and confidence intervals." BMJ: British Medical Journal 312.7038 (1996): 1079.

“how to choose different methods for analysis in reality work is still be in the controversial, due to the non-parameter test with less efficient as compared to parametric test”: It is controversial for a higher amount of samples in each group, not when groups have only 3 samples.

Response: Thanks for your question. We agree with your opinion.

“the non-parameter test with less efficient as compared to parametric test”: This is not an excuse to run away from the most adequate tests in these cases (very small number of samples in each group), which are non-parametric tests, as (already mentioned above) normality cannot be properly verified. That is why I wrote in my previous review that the authors are facing a big issue here, as nonparametric tests have less power to begin with and it’s a double whammy when you add a small sample size on top of that. But you cannot run away from that when there are only 3 samples in each group. Then, I will give one more chance to the authors to address this issue.

Response: Thanks for your insightful explanations and kindness. We have corrected all the original parametric analysis into nonparametric tests. The Mann-Whitney U Test and Kruskal-Wallis Test were employed to analyze the differences in Micro-CT and IHC results, respectively. The pairwise comparisons were used as the post hoc test of Kruskal-Wallis Test, the significance was also adjusted by the Bonferroni method, p < 0.05 was taken to indicate statistical significance between each group. You may find the modifications in 2.9 Statistical analysis; page 8, 263-269; page 18, 426-429, and corresponding correction also added to each figure. The supplementary information demonstrates the result which attained after a new nonparametric test. Last but not least, thanks for all of your questions and suggestions sincerely, they are critical and helpful for further improving the quality of our manuscript. We hope our answers and modifications could meet and satisfy your suggestions, thank you very much.

Round 3

Reviewer 3 Report

The manuscript now seems to be suitable for publication.